# Urinary Metabolites of Polycyclic Aromatic Hydrocarbons in Firefighters: A Systematic Review and Meta-Analysis

**DOI:** 10.3390/ijerph19148475

**Published:** 2022-07-11

**Authors:** Jooyeon Hwang, Chao Xu, Paul Grunsted, Robert J. Agnew, Tara R. Malone, Shari Clifton, Krista Thompson, Xin Xu

**Affiliations:** 1Department of Occupational and Environmental Health, Hudson College of Public Health, University of Oklahoma Health Sciences Center, Oklahoma City, OK 73104, USA; 2Department of Biostatistics and Epidemiology, Hudson College of Public Health, University of Oklahoma Health Sciences Center, Oklahoma City, OK 73104, USA; chao-xu@ouhsc.edu (C.X.); paul-grunsted@ouhsc.edu (P.G.); 3Fire Protection & Safety Engineering Technology Program, College of Engineering, Architecture and Technology, Oklahoma State University, Stillwater, OK 74078, USA; rob.agnew@okstate.edu; 4Department of Health Sciences Library and Information Management, Graduate College, University of Oklahoma Health Sciences Center, Oklahoma City, OK 73104, USA; tara-malone@ouhsc.edu (T.R.M.); shari-clifton@ouhsc.edu (S.C.); 5Department of Internal Medicine, University of Texas Southwestern Medical School, Dallas, TX 75390, USA; krista.thompson@utsouthwestern.edu; 6Shanghai Anti-Doping Laboratory, Shanghai University of Sport, Shanghai 200438, China; xuxin@sus.edu.cn

**Keywords:** biomarkers, biomonitoring, firefighter, hydroxylated polycyclic aromatic hydrocarbons (OHPAH), meta-analysis, occupational and environmental exposure, systematic review, urinary metabolites

## Abstract

Firefighters are intermittently exposed to complex, mixed pollutants in random settings. Of those pollutants, PAHs (polycyclic aromatic hydrocarbons) are the most commonly studied and best understood. PAH exposure can occur via multiple routes; therefore, the levels of hydroxylated metabolites of PAHs in urine have been used as a biomonitoring tool for risk assessment. We performed a systematic review and meta-analysis of the literature to estimate the levels of urinary hydroxylated PAH (OHPAH) among firefighters, determine risk attributions, and, finally, evaluate the scope of preventive efforts and their utility as diagnostic tools. The meta-regression confirmed increases in OHPAH concentrations after fire activities by up to 1.71-times (*p*-values: <0.0001). Samples collected at a time point of 2–4 h after a fire suppression showed a consistent, statistically significant pattern as compared with baseline samples. The National Fire Protection Association (NFPA) standard 1582 *Standard on Comprehensive Occupational Medical Program for Fire Departments* lists various health examinations, including a urinalysis for occupational chemical exposure if indicated and medical screening for cancers and cardiovascular diseases. Biomonitoring is a valuable screening tool for assessing occupational exposure and the results of this meta-analysis support their inclusion in regular health screenings for firefighters.

## 1. Introduction

Firefighters comprise one of the most challenging occupational groups to study using a traditional exposure assessment. Due to their unique working environment, the dimensions of their exposure (e.g., concentration, duration, and frequency) cannot be studied thoroughly. Specifically, firefighters are intermittently exposed to complexities of mixed pollutants in random settings during each on-call shift. Of those pollutants, PAHs (polycyclic aromatic hydrocarbons) are the most studied and best understood carcinogenic substances produced during firefighting activities. As previously reviewed [1], the possible routes of exposure for PAHs include not only inhalation but also dermal absorption and ingestion. When exposure occurs via multiple routes, an integrated approach for estimating systemic dose, called “biomonitoring”, is generally used. For example, urinary hydroxylated metabolites of PAHs have been used as a biomonitoring tool for the risk assessment of multi-route PAH exposure [2,3,4].

PAHs are absorbed into the human body through the respiratory tract, the skin, and the digestive tract, then disseminated to the lymph nodes, circulated in the blood, and metabolized in the liver and kidney [5]. Predominant PAHs are excreted in bile and urine, as their hydroxylated PAH (OHPAH) metabolites conjugate to water-soluble glucuronic acid and sulfate [6,7,8]. Other than bile and urine, the elimination routes of OHPAH metabolites in humans are feces and breast milk, yet the most commonly studied biospecimen is urine [3]. OHPAH compounds of four rings or more with higher molecular weights (MW) are mostly excreted in feces because their metabolism is more complex, whereas OHPAH compounds with lower MW tend to be excreted in urine because the hydroxyl metabolizes [2,3,4,9].

A recent systematic review covered a broad scope of firefighter biomonitoring studies that assessed occupational exposures to volatile organic compounds (VOCs) and semi-VOCs, including PAH and metals [10]. Another review study focused on reviewing firefighter urinary OHPAH metabolites as biomarkers after fire suppression [11], yet none explored the measured urinary OHPAH levels. We performed a systematic review and meta-analysis of quantitative synthesis of the evidence from the scarce literature to estimate the elevated levels of urinary OHPAH among firefighters, evaluate potential risk attributions, assess the scope of preventive efforts, and determine their utility as a diagnostic tool.

## 2. Materials and Methods

### 2.1. Search Strategy

We followed the Preferred Reporting Items for Systematic Reviews and Meta-Analyses (PRISMA) guidelines for the systematic review [12]. To identify publications, we searched the following databases: Embase (Classic + Embase OvidSP); MEDLINE (Epub Ahead of Print, In-Process, In-Data Review and Other Non-Indexed Citations and Daily OvidSP); Scopus; and Web of Science Core Collection (Figure 1). As the amount and quality of measurement data are limited, we did not include grey literature such as technical reports or those from the International Agency for Research on Cancer (IARC), conference proceedings, or theses/dissertations. We customized our search strategy by incorporating controlled vocabulary terms and/or keywords designed to retrieve literature relevant to the concepts ‘firefighters’ ‘urinary metabolites’, and ‘hydroxylated polycyclic aromatic hydrocarbons (OHPAH)’. The searches were conducted in May–June 2021. The detailed search strategies that reflect all terms and special features (e.g., limits, explode, focus, etc.) for each database are listed in the Appendix A. Search results from each database were imported into Covidence™ and were then screened.

### 2.2. Inclusion and Exclusion Criteria

We included original studies of urinary OHPAH metabolites from firefighters who were involved in firefighting activities. After screening against the title and abstract and assessing the eligibility of the full text, articles were excluded due to one or more of the following reasons:Different study scope (e.g., mathematical model, epidemiological study, animal model, chemical analytical approach);World Trade Center (WTC) cohort population;Non-firefighting activities (e.g., at the fire station, emergency medical services);Different biospecimens (e.g., serum);Different metabolites (e.g., Per- and Polyfluoroalkyl Substances (PFAS));Systematic review paper on biomonitoring including PAH;Editorial, case report, textbook, newsletters (e.g., science selection in a journal);Non-original study (e.g., companion, follow-up);Articles in non-English (e.g., German, Russian).

### 2.3. Data Extraction

We extracted descriptive characteristics from each article, including OHPAH analytical equipment; type of fire activity, either emergency or non-emergency (live training for structural and prescribed burn for wildfire); the number of participants; and type of biospecimen collections (Table 1). Each OHPAH metabolite and its relevant confounders, such as smoking status and grilled food consumption, were also extracted. These major recognized risk factors were restricted to dichotomous indices (yes/no). For the case of food intake, if indicated as no, the study followed the specific guidelines that the participants had no grilled, barbequed, or charcoaled food during study enrollment before urine collection. When needed, the metabolite values were estimated to the nearest decimal place based on a box plot, bar graph, or another figure. A third of the data points in the extracted studies were not normalized with the urinary creatinine levels. The consideration of adjusted creatinine is essential for OHPAH metabolites because there are possible diuresis-related effects depending on an individual body’s hydration at the time of urine collection. To minimize these biological effects, creatinine is measured in urine as an indicator of kidney function and protein intake [6,13]. When the studies reported the PAH metabolites without creatinine adjustments, we applied the conversion factor of 130.4 mg/dL, which is the average creatinine level derived from the 22,245 participants making up the study population in the 3rd National Health and Nutrition Examination Survey (NHANES III) [14]. The unit used for urinary PAH metabolites was ng/g—creatinine. Like our earlier meta-analysis of pre- and post- fire activity [1], we applied the concept of fold changes (FC). The FC represents a ratio of OHPAH metabolites mean between exposed and non-exposed participants of the fire activities, rather than a comparison of firefighters and a control group, such as the general population and/or a selected population without occupational exposure.

### 2.4. Statistical Analysis

The purpose of this statistical analysis was to conduct a random-effects meta-analysis on results from the literature to determine the influence of fire activity on the levels of OHPAH metabolites. Different study designs, methods of analysis, equipment and procedures, exposure scenarios, geographical settings, and emergency settings can cause significant variation and heterogeneity in overall effect estimates. Therefore, as the studies had heterogeneous characteristics, we used the generic inverse variance method [35] for a random-effects meta-analysis, which was implemented by the R package *meta* and *metafor.* The heterogeneity was investigated using the I^2^ statistic and Q-test. Missing values for the arithmetic mean (AM), geometric mean (GM), and standard error (SE) were replaced using an imputation method based on median and range. If a missing value could not be replaced, it was excluded from the meta-analysis. Two multi-regression models were applied to explore moderator effects of variables, such as the sampling time duration and sampling time point after exposure, with the following exposure-related metrics: grouped analytes and molecular weights. The moderator effect was estimated for the hypothesis of no effects using the SE and test *p*-value. To mitigate publication bias, we visually inspected the funnel plot (Appendix A), which maps effect sizes against their standard errors or precisions. The significance level was 0.05. All analyses reported here were conducted using R 4.0.5 (R Core Team, Vienna, Austria).

## 3. Results

A combined initial search yielded a total of 1340 articles (Figure 1). After removing duplicates (*n* = 775) and screening the titles and abstracts (*n* = 515), we assessed 50 full-text articles. Based on the selection criteria, 27 studies were included in the meta-analysis (Table 1). Eighteen (18) of those studies involved structural fires, while nine involved wildfires. Furthermore, twice as many studies were conducted at non-emergency fires (*n* = 18, live training and prescribed burns) than emergency fires (*n* = 9). Only one study [25] reported the results for emergency and live training separately in the same article. Although we did not specify a range of years, all of the reviewed studies were published in 1997 or later. Specifically, all except four studies were published after 2010. Since structural and synthetic materials constantly evolve, the characteristics of exposures at fires reported in the earlier publications probably do not reflect current exposure circumstances. Concerning the extracted evidence, each study used a different analytical method, which was presumably carried out using accessible laboratory equipment. These differences would warrant the development of standardizing analytical method.

The number of recorded data points was insufficient to perform the meta-analysis for each analyte. Moreover, PYR and ACE have the prefix number 1, indicating that they are non-congener metabolites related to fire emission exposure. In contrast, FLU has four (1, 2, 3, and 9) congeners. To avoid bias towards an individual analyte, as well as to increase the number of data points, we grouped analytes (e.g., OHFLU = 1FLU + 2FLU + 3FLU + 9FLU). Some studies directly reported OHPAH concentrations, which we included in the sum of all five metabolites. As shown in Table 2, the weights for fold change (FC) of the grouped metabolites increased significantly, from 1.35-times for OHPAH to 1.71-times for OHFLU, after a fire activity. Only three records were reported for OHACE and, thus, the *p*-value (1.00) across analyses was invalid. Similar to the analytes, the molecular weight (MW) was averaged to understand the impact of the time spent at the fire activity.

According to Table 3, a longer time spent (>30 min) showed a statistically significant increase in the FC, up to 1.93, with one exception. The exception of a shorter time spent at a fire activity showed a decrease in the FC at higher MW, but this was statistically significant (*p*-value < 0.0001).

As shown in Table 4, four sampling time point categories (baseline t = 0; 0–2 h, or t ≤ 2; 2–4 h, or t = 2–4; >4 h, or t ≥ 4) were used to identify the time window showing the best urinary OHPAH sample collection after exposure. Two grouped analytes (OHNAP and OHPHE) had values of FC < 1.0, indicating that weight is lower at post-fire t < 2 than at pre-fire t = 0. Yet, there was no statistically significant difference between t < 2 and t = 0. All weights for FC were greater than 1 when the sampling time point was 2–4 h after the fire activity. Except for OHNAP, all grouped analytes showed significant differences between t = 2–4 and t = 0 (*p*-value < 0.05). When the sampling time point was greater than 4 h from the baseline, FC was less than 1 for OHFLU, OHNAP, and OHPHE. As Table 5 shows, structural firefighters had a statistically higher weight of urinary OHPAH metabolites, up to 11.11-times for OHPYR, than those of wildland firefighters (*p*-values < 0.05).

In about half of the studies (*n* = 14), the participants were not permitted to eat grilled food for the duration of enrollment, which varied from 12 h to a week before the study started. Similarly, half of the studies (*n* = 13) excluded smokers from participation. To avoid possible interaction effects, we analyzed OHPAH levels with grilled food consumption (restricted grilled food versus no restriction) and tobacco exposure (recruited smoker versus non-smoker) as variables. Our meta-regression provided mixed results as the type/amount of diet and smoke greatly varies between firefighters. The levels of OHNAP and OHPYR were significantly higher in the studies with participants who ate grilled food (*p*-values < 0.0001), while OHPHE and OHFLU were not (*p*-values = 0.56 and 0.22, respectively). The results for tobacco consumption were analogous to those for diet (data not shown).

## 4. Discussion

### 4.1. Metabolite Quantification

Only a few of the hydroxylated metabolites of PAHs in the firefighter urine were quantified in the articles we reviewed. Furthermore, hydroxylated metabolites result from only one of many potential metabolic pathways. In general, PAH analytes produce OHPAH metabolites, but not always. For example, 3-BaP is one of the metabolites of benzo(a)pyrene. Similarly, 1-NAP, which is frequently used to assess exposure to naphthalene, is a metabolite of not only naphthalene but also the insecticide carbaryl [36]. Because only certain PAHs produce a number of the urinary metabolites of the OHPAH isomer [7], the correlation between PAH (e.g., naphthalene) and OHPAH (e.g., 1-NAP) was moderate (r = 0.57) [9]. Although our review excluded WTC firefighters, Edelman (2003) [37] measured seven additional urinary metabolites, 1-hydroxybenzo[a]anthracene, 1-hydroxybenzo[c]phenanthrene, 2-hydroxybenzo[c]phenanthrene, 3-hydroxybenzo[a]anthracene, 3-hydroxybenzo[c]phenanthrene, 3-hydoxychrysene, and 3-hydroxyfluoranthrene, in their study cohort in response to WTC. Due to hydroxyl metabolism, there have been no urinary studies of OHPAH with five or six rings, including benzo[e]pyrene and benzanthrone, as mostly found in feces. In 9 of the 27 studies we reviewed, 1-PYR, which is the biomarker most commonly used to monitor complex carcinogenic PAH exposure [2,9], was the sole biomarker analyzed. However, according to our meta-analysis, there was a significant increase in all hydroxylated PAH levels except OHPYR in firefighter urine samples post-fire (see Table 2). In particular, Banks et al. (2021) and Anderson et al. (2018) [9,33] found that firefighting activities did not increase the 1-PYR concentration in urine, although the levels of 1-PYR and PYR on the skin were correlated [32]. While most studies investigated a single analyte using 1-PYR, two of the studies [13,15] we reviewed reported on up to 14 analytes of urinary OHPAH. This inconsistency warrants that the quantification analysis of urinary OHPAH be standardized, and that the laboratory process be part of the biomonitoring program for firefighters. In particular, Gill et al. (2019) [6] emphasized that standardized protocols, such as adding enzyme reagents for urine processing, will reduce variability between laboratories.

### 4.2. Sampling Time Windows

Previous studies of urine biospecimens have explored the best time points for sampling post-fire. Determining when to sample is especially challenging in the case of firefighters, who face multiple routes of PAH exposure that produce urinary OHPAH with different biological half-lives [22]. The two main routes of exposure are dermal and respiratory, both of which increase PAH metabolites. However, it is unclear which route is the main driver. Further, no definitive answer has been given concerning the pathway of PAH exposure resulting from firefighting. These uncertainties arise because the pharmacokinetics of metabolites differ by individual OHPAH analyte and individual firefighter, as well as the specific exposure route and assigned task [28,38]. Firefighters are situated in a unique working environment, fire suppression, and depend heavily on personal protective equipment, including turnout gear [39]. The metabolites resulting from dermal exposure to contaminated gear tend to have a longer half-life than the metabolites resulting from inhalation and excreted in urine, which possibly explains the bi-phase excretion pattern [34].

In the reviewed articles, most studies set multiple short-term sampling time points, such as during the controlled burn, immediately after the burn, three hours post-exposure, and six hours post-exposure. The study with the most sampling time points had nine times, ranging from immediately after to 18 h post-exposure [13]. The longest time point for sampling was two weeks post-exposure [6]. Another group [26] targeted to maximize the level of OHNAP; thus, they collected urine samples 2–4 h after exposure. Yet, as they pointed out, this sampling method likely underestimates the post-fire level of urinary OHPAH, which has a longer half-life until elimination. A sampling time of three hours or less may capture the peak excretion of many OHPAH, thus, serving as a short-term biomarker [15,23,25]. In this meta-analysis, the FC (fold change), which is the ratio of a sample collected at 2–4 h post-exposure to the baseline, was consistently greater than 1, reflecting that there are no dilution effects in this time window (see Table 4). Conversely, the FC for the samples collected >4 h post-exposure remained high for OHPYR at 9.17, perhaps because the half-life of PYR is longer than those of the other urinary OHPAH [26]. This finding can also be ascribed to accumulation effects. Cherry et al. (2021) [34] report that the level of 1-PYR post-shift did not recover to the pre-shift level overnight due to accumulated exposures. Rossbach et al. (2020) [13] estimated that the elimination half-life of individual PAH metabolites, including 1-PYR, is 7.6 h. Similarly, they estimate that the maximal time for urinary excretion of 1-PYR is 5.5–6 h [15,38].

### 4.3. Structural versus Wildland Fires

PAHs, which are ubiquitous during fire suppression, are produced by the incomplete combustion of both natural and anthropogenic sources [4]. Occupational exposure assessments of PAH have focused heavily on the latter, structural fires, due to growing concerns with the synthetic materials used to fill modern structures. In contrast, exposure to PAHs occurring due to wildland fires are seldom specifically addressed. In our review, we included studies of both structural and wildland firefighters.

A comparison of PAH metabolites produced by structural as opposed to wildland fires (see Table 5) is problematic, mainly due to different external working environments and respiratory protection practices. For example, the duration of fire suppression is significantly different between the two types of fire. Structural firefighters perform their tasks in approximately 30 min [40], while wildland firefighters spend extended periods in wildfires, averaging from 8 to 13 h [41,42]. Regarding respiratory protection, structural firefighters usually wear an SCBA (Self-Contained Breathing Apparatus) [43], which has the highest assigned protection factor (APF = 10,000) of all types of respiratory protection equipment. However, wildland firefighters wear NIOSH-approved N95 filtering facepiece respirators [44], cotton bandanas (Nomex^®^ shrouds), or often nothing. None of these latter options supply fresh air or oxygen; none protect against gas-phase contaminants; and only the N95 respirator protects against particulate-phase contaminants [41]. Thus, the main route of exposure for wildland firefighters may be inhalation [18]. In contrast, structural firefighters may be exposed to PAH while waiting their turn at live fire training [32] or at an emergency fire adjacent to the burning structure, often without SCBA. The consequence of such short exposure is minimal; thus, the main route of exposure for structural firefighters may be dermal from contaminated gear [3,18].

### 4.4. Recommendations

The time between end-of-fire suppression at a fireground and return to station is critical, as chemicals may be absorbed either through dermal exposure or off-gassing from contaminated gear. Correspondingly, NFPA 1851 [45] has new language that highlights the time sensitivity of preliminary exposure reduction at a fireground or emergency scene. This change is in line with numerous studies that have reported that engaging in hygiene practices immediately after a fire activity decreases exposure to PAHs. PAHs activate aryl hydrocarbon receptors (AhR), resulting in inflammatory, oxidative, and genotoxic effects [46]. One study found that firefighters who used a baby wipe removed residuals, including AhR active compounds, from their skin [24].

At a fireground, intervention practices can include gross decontamination of gear with soap before doffing as well as skin decontamination [34]. As part of field decontamination, we also recommend that firefighters rinse debris and mucus from their nasal cavities using a saline nasal spray. However, this intervention has not yet been investigated and should be further studied. Before returning to the station, they should collect contaminated gear and store it in an airtight container in a fire truck [47]. Upon arriving at the station, an extractor should be used to wash the gear while the firefighters immediately take showers [47]. An infrared sauna comparison of pre- and post-live fire training found that OHPAH in urine was reduced by 43% by stimulating an increase in sweat [25].

NFPA 1582 [48] provides a detailed list of health exams, including a blood test, urinalysis, and medical screening for cancers and cardiovascular diseases. In the most recent version, the guidelines for screening nine types of cancer have been updated. Based on the rising rates of cancer and research that shows the health benefits of prevention methods and early treatment, we believe all firefighters should undergo an annual health assessment. Yet, this standard is not mandatory and often not feasible for many rural volunteer fire departments due to logistical and budgetary constraints. Alternatively, the collection of urine specimens can be a valuable tool for assessing internal biological indicators of exposure dosage. Based on the stage and prognosis of a malignancy in the urinary tract, urine has been used as a valid biomarker for prostate [49,50], bladder [51], and renal cell carcinoma [52]. Urine collection is non-invasive, easy, and safe to obtain from firefighters and allows for repetitive measurement, which is suitable for longitudinal studies.

### 4.5. Limitations

The health risks posed to firefighters from occupational exposure to contaminants may differ based on meteorological parameters, type of fuel (vegetation or building materials), size of the burn, length of work shift, or exposure to diesel exhaust, particulate matters, and volatile organic compounds. We only considered a few of these risk factors in our meta-analysis. In addition, firefighters perform various tasks on non-burn workdays, including emergency medical service (EMS), administrative duties, required training, including respiratory protection programs, engine maintenance, etc. There is a moderate to strong correlation between airborne total PAHs and urinary total OHPAH at a fire station [53], which raises the possibility of additional exposure to contaminants while firefighters are on duty, regardless of whether they are directly engaged in fire activities. Our meta-analysis was limited to fire activities, not to firefighters’ duties as a whole. Another limitation is regarding the OHPAH molecular weight (MW). The low MW of PAH, which predominantly exists in gaseous form and is highly volatile, may lead to underestimates of personal exposure levels [53]. Simultaneously, due to their low MW and high volatility, PAHs, such as NAP (MW: 144.17 g/mol), may penetrate the protective layers of turnout gear [25], which increases the level of PAH in urine. Keir et al. (2017) [21] presented empirical evidence showing that inhalation is not the main route of exposure for NAP. Instead, the concentration of NAP is the highest PAH in personal air samples and the lowest PAH metabolites in urinary samples. One possible reason is that turnout gear, the main protective personal equipment used by firefighters, protects from fire, not from volatile chemicals, such as low-MW PAHs [13]. Furthermore, a strong inverse correlation exists between OHPAH and MW. The higher the level of urinary OHPAH in firefighters, the lower the MW of un-metabolized congener compounds in their urine [16,53,54]. Consistent with this correlation, the overall mean of OHNAP was highest in our meta-regression analysis. In contrast, PAHs with high MW are distributed in the gas or particulate phases, typically bound to particles, and have low volatility [3]. As particle-bound metabolites of PAH are eliminated in feces, the concentration of high-MW PAHs is low or non-existent in urine. For example, Oliveira et al. (2016) [3] did not find any 3-hydroxybenzo[a]pyrene (3-BaP, MW: 268.3 g/mol) in the urine of either non-exposed or exposed firefighters. Metabolites of other high-MW PAHs, such as 1-hydroxypyrene (1-PYR) and 3-hydroxyfluoranthene (3-FLU), were detected in less than half of the firefighters’ urine [7]. Hence, urine biospecimens may not fully capture PAH toxicity because the carcinogenic metabolites of PAHs have high MW. Biomonitoring both urine and fecal samples of firefighters will cover these gaps.

## 5. Conclusions

Firefighters are exposed to an extensive list of contaminants through multiple routes and depending on tasks specific to each unique fire. While the main routes of exposure for structural firefighters may be dermal from contaminated gear, the main routes of exposure for wildland firefighters may be through inhalation from inadequate respiratory protection practices against contaminants. Thus, biomonitoring the internal doses of urinary metabolites has the potential to be a powerful tool for assessing firefighters’ exposure due to fire suppression. OHPAH is metabolized and excreted through urine; the collection of urine biospecimens from firefighters 2–4 h after the fire activities end will address gaps in the levels of PAH carcinogens. As a part of the biomonitoring program for firefighters, urinary OHPAH needs to be standardized for the quantification analysis and harmonization of laboratory processes. Our recommendation for firefighters is to rinse debris and mucus from their nasal cavities as part of their field decontamination. Further, firefighters should have the benefit of an annual health assessment for occupational exposure, including urinalysis, if indicated, and preventive care and medical screening for cancers and cardiovascular diseases.

## Figures and Tables

**Figure 1 ijerph-19-08475-f001:**
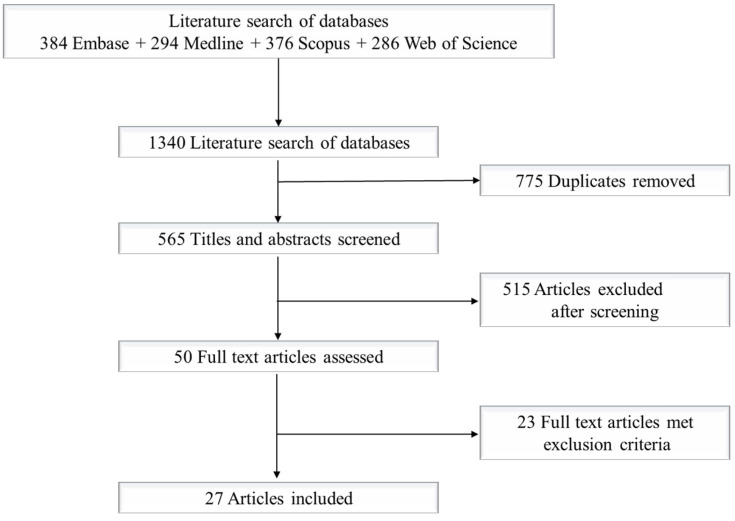
PRISMA flow chart of the literature screening and selection processes.

**Table 1 ijerph-19-08475-t001:** Summary of study characteristics in evaluations of urinary OHPAH metabolites in firefighters.

Study ID	Author/ Year	Country/ State or Region	Analytical Equipment	Type of Fire	Class of Fire	No. of Participants	Male No. (%)	Other Biospecimens Collected	Smoke (Y/N)	Grilled Food (Y/N)
150	Oliveira 2016 [3]	Portugal/ Trás-os-Montes	LC-FLD	Wildfire	Emergency	153	NR ^1^	– ^2^	N ^3^	N
161	Oliveira 2017 [4]	Portugal/Bragança	LC-FLD	Wildfire	Emergency	108	NR	–	Y	N
212	Fent 2019 [15]	U.S./Illinois	HPLC-MS	Structural	Live training	34	31 (91.2)	Breath	N	N
153	Adetona 2017 [16]	U.S./S. Carolina	HPLC-MS	Wildfire	Prescribed burns	19	17 (89.5)	–	Y	Y
142	Fernanado 2016 [7]	Canada/Ontario	GC-MS/MS	Structural	Live training	28	24 (85.7)	–	N	N
203	Cherry 2019 [17]	Canada/Alberta	LC-MS/MS	Wildfire	Emergency	172	162 (94.2)	–	Y	Y
221	Adetona 2019 [18]	U.S./S. Carolina	HPLC-FLD	Wildfire	Prescribed burns	12	9 (75)	–	Y	Y
115	Fent 2014 [19]	U.S./Illinois	ELISA	Structural	Live training	18	18 (100)	Breath	N	N
72	Laitinen 2010 [20]	Finland/Kuopio; France/Paris	LC-FLD; GC-ECD	Structural	Live training	16	NR	–	N	NR
183	Wingfors 2018 [2]	Sweden/Sando	GC-MS/MS	Structural	Live training	20	NR	–	N	N
202	Gill 2019 [6]	Canada/Alberta	LC-MS/MS; GC-HRMS ^4^	Wildfire	Emergency	42	42 (100)	–	Y	Y
173	Keir 2017 [21]	Canada/Ontario	GC-MS/MS	Structural	Emergency	44	44 (100)	–	N	N
88	Laitinen 2012 [22]	Finland/Kuopio	LC-FLD; GC-ECD	Structural	Live training	13	NR	–	N	NR
244	Fent 2020 [23]	U.S./Illinois	ELISA; HPLC-MS/MS	Structural	Live training	41	37 (90.2)	Breath	N	N
237	Rossbach 2020 [13]	Germany/Frankfurt & Main	GC-MS/MS	Structural	Live training	6	NR	–	Y	Y
229	Beitel 2020 [24]	U.S./Arizona	GC-MS/MS	Structural	Live training	11	11 (100)	–	N	N
271	Burgess 2020 [25]	U.S./Arizona	GC-MS	Structural	Emergency	242	NR	Blood, Buccal cells	Y	Y
					Live training	24	NR	–	N	N
291	Hoppe-Jones 2021 [26]	U.S./Arizona	GC-MS	Structural	Emergency	242	NR	Blood, Buccal cells	Y	Y
280	Banks 2021 [9]	Australia/Queensland	GC-MS/MS	Structural	Live training	26	25 (96.2)	–	N	Y
36	Caux 2002 [27]	Canada/Toronto	HPLC-UV	Structural	Emergency	43	NR	–	Y	Y
151	Oliveira 2020 [28]	Portugal/Bragança	LC-FLD	Wildfire	Emergency	171	NR	Blood, Cardio–respiratory	Y	N
22	Moen 1997 [29]	Norway/Bergen	HPLC	Structural	Live training	13	NR	–	Y	N
20	Feunekes 1997 [30]	Netherlands/Den Helder	HPLC	Structural	Live training	47	NR	–	Y	NR
65	Robinson 2008 [31]	U.S./Arizona	HPLC-FLD	Wildfire	Prescribed burns	21	NR	Lung function	Y	Y
179	Andersen 2018 (a) [32]	Denmark/Copenhagen	HPLC-FLD	Structural	Live training	53	41 (77.4)	Blood, Skin, Lung function	N	Y
190	Andersen 2018 (b) [33]	Denmark/Copenhagen	HPLC-FLD	Structural	Emergency	22	22 (100)	Blood, Skin, Lung function	Y	Y
261	Cherry 2021 [34]	Canada/Alberta or British Columbia	LC-MS/MS	Wildfire	Prescribed burns	86	66 (76.7)	–	N	Y

^1^ NR: Not recorded; ^2^ Urine only; ^3^ All participants were non-smokers; ^4^ Meta-analysis used data by GC-HRMS.

**Table 2 ijerph-19-08475-t002:** Pre/post comparison of hydroxylated PAH (OHPAH) collected from urine samples (unit: ng/g—creatinine) by grouped analytes. Bold *p*-value indicates a statistically significant difference in urinary OHPAH levels after the fire activity.

Grouped Analyte *	No. Records	Post-Pre Fire Activity		Fold Change (Post/Pre)	*p*-Value
		Mean	SE		
OHFLU	122	6.3	1.17	1.71	**<0.0001**
OHNAP	106	35.6	2.37	1.57	**<0.0001**
OHPHE	103	2.3	0.19	1.58	**<0.0001**
OHPYR	77	0.3	0.02	0.40	**<0.0001**
OHPAH	451	1.0	0.04	1.35	**<0.0001**

* OHFLU (hydroxyfluorenes) = 1FLU (1-hydroxyfluorene) + 2FLU (2-hydroxyfluorene) + 3FLU (3-hydroxyfluorene) + 9FLU (9-hydroxyfluorene); OHNAP (hydroxynaphthalenes) = 1NAP (1-hydroxynaphthalene) + 2NAP (2-hydroxynaphthalene); OHPHE (hydroxyphenanthrenes) = 1PHE (1-hydroxyphenanthrene) + 2PHE (2-hydroxyphenanthrene) + 3PHE (3-hydroxyphenanthrene) + 4PHE (4-hydroxyphenanthrene) + 9PHE (9-hydroxyphenanthrene); OHPYR (hydroxypyrenes) = 1PYR (1-hydroxypyrene); OHPAH = sum of all metabolites, plus direct report of OHPAH from studies.

**Table 3 ijerph-19-08475-t003:** Duration of exposure comparison of hydroxylated PAH (OHPAH) collected from urine samples (unit: ng/g—creatinine) by molecular weight. Bold *p*-value indicates a statistically significant difference in urinary OHPAH levels by time spent at fire activity.

Molecular Weight *	No. Records	>30 min–≤30 min		Fold Change (>30 min/≤30 min)	*p*-Value
		Mean	SE		
144.17	50	23.1	6.63	1.60	**0.0005**
182.22	65	2.9	0.72	1.93	**<0.0001**
194.23	48	0.4	0.12	1.44	**0.0003**
218.25	67	−0.4	0.02	0.24	**<0.0001**

* 144.17 g/mol = 1NAP, 2NAP; 182.22 g/mol = 1FLU, 2FLU, 3FLU, 9FLU; 194.23 g/mol = 1PHE, 2PHE, 3PHE, 4PHE, 9PHE; 218.25 g/mol: 1PYR.

**Table 4 ijerph-19-08475-t004:** Three sampling time windows (t < 2, t = 2–4, t > 4) and one baseline (t = 0) of OHPAH were collected from urine samples (unit: ng/g—creatinine) by grouped analytes. Bold *p*-value indicates statistically significant difference in urinary OHPAH levels at a sampling time point after and before fire activity.

Grouped Analyte *	No. Records	[t ≤ 2]– [t = 0] h Mean	SE	Fold Change	*p*-Value	[t = 2–4]–[t = 0] h Mean	SE	Fold Change	*p*-Value	[t => 4]– [t = 0] h Mean	SE	Fold Change	*p*-Value
OHFLU	69	7.2	3.10	2.01	**0.0210**	3.3	1.66	1.47	**0.0475**	−2.2	2.25	0.69	0.3375
OHNAP	65	−26.5	33.18	0.76	0.4245	21.1	16.35	1.19	0.1978	−85.2	21.87	0.24	**<0.0001**
OHPHE	58	−1.0	1.91	0.82	0.5830	5.0	1.30	1.85	**0.0001**	−3.7	1.63	0.37	**0.0226**
OHPYR	40	1.9	0.22	23.69	**<0.0001**	2.7	0.07	33.02	**<0.0001**	0.7	0.02	9.17	**<0.0001**
OHPAH	253	0.7	0.07	2.24	**<0.0001**	2.3	0.06	4.88	**<0.0001**	1.4	0.04	3.31	**<0.0001**

* OHFLU = 1FLU + 2FLU + 3FLU + 9FLU; OHNAP = 1NAP + 2NAP; OHPHE= 1PHE + 2PHE + 3PHE + 4PHE + 9PHE; OHPYR = 1PYR; OHPAH = sum of all metabolites, plus direct report of OHPAH from studies.

**Table 5 ijerph-19-08475-t005:** Structural and wildland fire comparison of hydroxylated PAH (OHPAH) collected from urine samples (unit: ng/g—creatinine) by grouped analytes. Bold *p*-value indicates a statistically significant difference in urinary OHPAH levels between structural and wildland fire activity.

Grouped Analyte *	No. Records	(Wildfire-Structural)		Fold Change (Structural/Wildfire)	*p*-Value
		Mean	SE		
OHFLU	122	3.3	1.42	2.08	**0.0195**
OHNAP	–	–	–	–	–
OHPHE	103	2.5	0.22	5.88	**<0.0001**
OHPYR	77	1.0	0.01	11.11	**<0.0001**
OHPAH	451	1.2	0.04	1.96	**<0.0001**

* OHFLU = 1FLU + 2FLU + 3FLU + 9FLU; OHNAP = 1NAP + 2NAP; OHPHE= 1PHE + 2PHE + 3PHE + 4PHE + 9PHE; OHPYR = 1PYR; OHPAH = sum of all metabolites, plus direct report of OHPAH from studies.

## Data Availability

Data compiled for this meta-analysis are available on request from the corresponding author. The R-code is deposited on GitHub repository (https://github.com/PaulG-Epi/Urinary-Metabolites-of-PAH-in-Firefighters-Meta-analysis.git (accessed on 10 May 2022).

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
