# Peer review of "Urinary Metabolites of Polycyclic Aromatic Hydrocarbons in Firefighters: A Systematic Review and Meta-Analysis"

_ijerph, 2022, doi:10.3390/ijerph19148475_

Round 1

Reviewer 1 Report

Line 33 - There is a little confusion about some concepts. Then, I suggest that the authors use the expression biomonitoring instead "urinalysis" as a tool for assessing occupational exposure, but not assessing internal biological indicators of PAH.

Line 168 - write the names of the metabolites in full before the abbreviation presentation on the tables

Author Response

We greatly appreciate their useful comments, which have markedly improved the manuscript. The revised manuscript synthesized all comments from the reviews. As each pointed out, given track changes are our responses to each comment and our subsequent revision. Once again, thank you for your review.

1. Line 33 - There is a little confusion about some concepts. Then, I suggest that the authors use the expression biomonitoring instead "urinalysis" as a tool for assessing occupational exposure, but not assessing internal biological indicators of PAH.

We have revised in line 33

2. Line 168 - write the names of the metabolites in full before the abbreviation presentation on the tables

We have included full names of metabolites in line 179.

Reviewer 2 Report

Authors Hwang et al. present an interesting systematic review and meta-analysis of OHPAH in firefighters. The study is well-reasoned and considers many important factors in such occupational exposures including type of fire, class of fire, smoke inhalation, length of exposure, consumption of grilled foods, etc. As opposed to previous systematic reviews on the topic, the current meta-analysis explores measured urinary OHPAH levels and therefore provides a quantitative synthesis currently not available in the literature. This is a good paper deserving of publication in IJERPH following minor revisions:

1.     Alphabetize the keywords.

2.     Lines 56 and 57: text should say ‘glucuronic acid and sulfate’. The ‘acid’ is missing.

3.     Line 58: Please change ‘biomarker’ to ‘biospecimen’. Urine is not a biomarker in and of itself in this case.

4.     Table 1 is listed at the very end, after all other tables. Yet, it is first mentioned on line 108. Please move Table 1 up to be immediately after the paragraph of its first mention.

5.     Section 4.2 of the Discussion should be moved to the limitations section, since it is a discussion of the limitations of primarily low MW compounds found in urine. It is a great discussion of this limitation but should not be its own section. So, in total, there should be five sections.

6.     Section 4.4 of the Discussion, beginning on line 305, should (in addition to being relabeled section 4.3 after moving section 4.2 to the limitations) be moved up to the previous page. As is formatted, there is a large gap between ‘Sampling Time Windows’ and ‘Structural Versus Wildfires’. Also, italicize the subsection title for section 4.4.

7.     The conclusion is relatively scarce as is. Please make sure at least one statement summarizing each of the major findings of the review are contained in the conclusion. It’s currently underwhelming and lacks many of the very interesting associations discovered in your meta-analysis.

Many thanks to the authors for their work and for soliciting my review. I hope these suggestions improve the quality of their manuscript. I would be pleased to receive a revised draft for subsequent review.

Author Response

We greatly appreciate their many useful comments, which have markedly improved the manuscript. The revised manuscript synthesized all comments from the reviews. As each pointed out, given track changes are our responses to each comment and our subsequent revision. Once again, thank you for your review.

1. Alphabetize the keywords. We rearranged the keywords.

2. Lines 56 and 57: text should say ‘glucuronic acid and sulfate’. The ‘acid’ is missing. We included the missing word in line 57.

3. Line 58: Please change ‘biomarker’ to ‘biospecimen’. Urine is not a biomarker in and of itself in this case. We changed to biospecimen in line 58.

4. Table 1 is listed at the very end, after all other tables. Yet, it is first mentioned on line 108. Please move Table 1 up to be immediately after the paragraph of its first mention. Thank you for your suggestion. We moved Table 1 right after the paragraph in line 110.

5. Section 4.2 of the Discussion should be moved to the limitations section, since it is a discussion of the limitations of primarily low MW compounds found in urine. It is a great discussion of this limitation but should not be its own section. So, in total, there should be five sections. Thank you for your insights. We removed Section 4.2 and moved to Section 4.5 Limitations. Now, there are five sections in Discussion. 

6. Section 4.4 of the Discussion, beginning on line 305, should (in addition to being relabeled section 4.3 after moving section 4.2 to the limitations) be moved up to the previous page. As is formatted, there is a large gap between ‘Sampling Time Windows’ and ‘Structural Versus Wildfires’. Also, italicize the subsection title for section 4.4. Thank you for your suggestion. We accordingly reformatted those sections in the Discussion.

7. The conclusion is relatively scarce as is. Please make sure at least one statement summarizing each of the major findings of the review are contained in the conclusion. It’s currently underwhelming and lacks many of the very interesting associations discovered in your meta-analysis. We appreciate the suggestion. We summarized each of major finding in the Conclusion.

Reviewer 3 Report

The reviewed literature in this investigation is  acceptable. The cited references are acceptable.

(i) Strength of the ms: This manuscript s well written. It thus meets  criteria for publication. One major critique of the ms  is given here below.

(ii) A major critique to the review:

A major critique to this review is the limitation to oxy-PAHs of  up-to 4 benzenoid rings. Although oxy-metabolites of benzo(a)pyrene are mentioned in this manuscript, the reader is  left with questions on their existence. Thus, oxy-PAH metabolites of 5-benzonoid PAHs such as benzo(e)pyrene, and benzanthrone may need to be given mention in this manuscript. 

Without mention of such metabolites, then the manuscript will  not be recommended for publication.

(iii) Recommendation:  The reviewer is of the opinion that the ms may meet publication after the authors address the critique and question posed in (ii) above.

Author Response

We greatly appreciate their many useful comments, which have markedly improved the manuscript. The revised manuscript synthesized all comments from the reviews. As each pointed out, given track changes are our responses to each comment and our subsequent revision. Once again, thank you for your review.

(ii) A major critique to the review: A major critique to this review is the limitation to oxy-PAHs of  up-to 4 benzenoid rings. Although oxy-metabolites of benzo(a)pyrene are mentioned in this manuscript, the reader is  left with questions on their existence. Thus, oxy-PAH metabolites of 5-benzonoid PAHs such as benzo(e)pyrene, and benzanthrone may need to be given mention in this manuscript. Thank you for your suggestion. We have clarified the suggestion in lines 239-245.